# Review of Harmful Algal Blooms in the Coastal Mediterranean Sea, with a Focus on Greek Waters

Christina Tsikoti [1] and Savvas Genitsaris [2,*]

1 School of Humanities, Social Sciences and Economics, International Hellenic University, 57001 Thermi, Greece; c.tsikoti@ihu.edu.gr
2 Section of Ecology and Taxonomy, School of Biology, Zografou Campus, National and Kapodistrian University of Athens, 16784 Athens, Greece
* Correspondence: genitsar@biol.uoa.gr; Tel.: +30-210-7274249

**Abstract:** Anthropogenic marine eutrophication has been recognized as one of the major threats to aquatic ecosystem health. In recent years, eutrophication phenomena, prompted by global warming and population increase, have stimulated the proliferation of potentially harmful algal taxa resulting in the prevalence of frequent and intense harmful algal blooms (HABs) in coastal areas. Numerous coastal areas of the Mediterranean Sea (MS) are under environmental pressures arising from human activities that are driving ecosystem degradation and resulting in the increase of the supply of nutrient inputs. In this review, we aim to present the recent situation regarding the appearance of HABs in Mediterranean coastal areas linked to anthropogenic eutrophication, to highlight the features and particularities of the MS, and to summarize the harmful phytoplankton outbreaks along the length of coastal areas of many localities. Furthermore, we focus on HABs documented in Greek coastal areas according to the causative algal species, the period of occurrence, and the induced damage in human and ecosystem health. The occurrence of eutrophication-induced HAB incidents during the past two decades is emphasized.

**Keywords:** HABs; Mediterranean Sea; eutrophication; coastal; phytoplankton; toxin; ecosystem health; disruptive blooms

## 1. Introduction

Cultural (or anthropogenic) coastal marine eutrophication has been recognized as one of the major threats to the health of marine environments and can be an expression of ecosystem disturbance [1]. At the present time, > 50% of the world's population lives in urban areas, mostly located in coastal regions or regions strongly influenced by coastal systems [2]. The main drivers of coastal eutrophication are immediately connected to human activities, which increase the supply of nutrient inputs, especially the supply of nitrogen (N) and phosphorus (P). Drivers, such as river inputs, land runoff, and industrial or urban wastewater, are strongly related to the increase of the human population, industrialization, and intensification of agriculture [3,4]. As predicted, the increase in the global population will coincide with a rise in the rates of coastal urbanization. This fact, combined with the global rise of temperature and sea level, the intensified hydrological cycles, and the shift in wind patterns, would likely increase the risk of coastal eutrophication in the coming years [5].

During the last two decades, cultural eutrophication in combination with climate change has stimulated the prevalence, significance, and geographic extent of Harmful Algal Blooms (HABs), which include potentially toxic species [6]. HABs can be described as incidents where the assemblage of one or more different harmful algae reach abundances potentially harmful to other marine organisms, causing, for example, the kills of fish or/and shellfish, the accumulation of algal toxins in marine organisms, and the intoxication of human consumers via the consumption of shellfish [7]. HAB is a broad definition that

includes many phenomena. According to the report from Task Group 5 on Marine Strategy Framework Directive, there are three types of harmful algal blooms: (1) those caused by toxic algae (e.g., *Alexandrium*, *Dinophysis*, and *Pseudo-nitzschia*), which can cause shellfish toxicity even in low concentrations; (2) those caused by likely toxic algae (e.g., *Pseudo-nitzschia*); and (3) high-biomass blooms (e.g., *Karenia*, *Phaeocystis*, *Noctiluca*) that create problems due to the high biomass itself and the production of organic matter [8,9].

Of the $\geq$ 5000 phytoplankton species that have been described worldwide, nearly 300 species from most phylogenetic groups can create high-biomass red tides. Those species mainly belong to the dinoflagellates, diatoms, and haptophytes and are acknowledged to be toxic to fish, shellfish, marine mammals, and humans [10,11] (Table 1). However, there is no overall guideline to set the limits of harmful cell abundances in an algal bloom: the abundance of harmful algae (expressed as the number of cells) does not define the HAB—as some species are so toxic that their existence, even in rather low concentrations, may be harmful [7].

**Table 1.** Harmful Algal Bloom (HAB) categories and the main responsible algae for the formation of the bloom [7,12].

| | HAB Categories | Description | Responsible Algae |
|---|---|---|---|
| I | Water discoloration | HABs that principally cause harmless discoloration of seawater; result in decreased recreational value of affected area due to low water clarity; bloom can grow so dense that they cause oxygen depletion leading to death of fish and benthic invertebrates under extreme conditions in confined bays. | Dinoflagellates: *Noctiluca scintillans*, *Ceratium* spp., *Prorocentrum micans*, *Heterocapsa triquetra*, *Akashiwo sanguinea*, *Gonyaulax polygramma*, *Scrippsiella trochoidea*, *Peridinium quinquecorne* Euglenophytes: *Eutreptiella* spp. Haptophytes: *Phaeocystis* spp. Diatoms: *Skeletonema costatum*-complex |
| II | Toxin accumulation harmful to humans | HABs that produce toxins that accumulate in food chains causing gastrointestinal and neurological symptoms to humans and animals; such as Paralytic Shellfish Poisoning (PSP), Diarrhetic Shellfish Poisoning (DSP), Neurotoxic Shellfish Poisoning (NSP), Amnesic Shellfish Poisoning (ASP) and Ciguatera Fish Poisoning (CFP). | Dinoflagellates: *Alexandrium* spp., *Gymnodinium catenatum*, (PSP); *Dinophysis acuminata*, *Dinophysis* spp., *Prorocentrum lima* (DSP); *Karenia brevis*, *Karenia* spp. (NSP) *Gambierdiscus* spp. (CFP), *Coolia* spp. Diatoms: *Pseudo-nitzschia* spp. (ASP) |
| III | Toxin production harmful to wildlife | HABs usually non-toxic to humans but toxic to fish and invertebrates (especially in intensive aquaculture systems), e.g., by intoxication or by causing damage or clogging of the gills. | Dinoflagellates: *Alexandrium tamarense*-complex, *Gyrodinium aureolum*, *Karenia mikimotoi*, *Karlodinium micrum*, Haptophytes: *Chrysochromulina polylepis* Diatoms: *Prymnesium parvum*, *Prymnesium patelliferum* Raphidiophytes: *Heterosigma akashiwo*, *Chattonella antiqua* Pelagophytes: *Aureococcus anophagefferens* Cyanobacteria: *Nodularia spumigena* |
| IV | Production of aerosolized toxins | HABs that produce toxins toxic to humans, which are transferred by air in aerosols from the bloom area to the coast. | Dinoflagellates: *Pfiesteria* spp., *Karenia brevis*, *Ostreopsis* spp. |

It is well-established that there has been an increase in the susceptibility of coastal environments to HABs [6,13]. Although links between anthropogenic eutrophication and HABs in coastal areas worldwide have been recognized, these linkages are not always clear since the processes through which human activities lead to nutrient enrichment are rather complicated [14]. At a global level, the coastal areas that are documented to be highly affected by HABs include the Gulf of Mexico and the Chesapeake Bay, USA, coasts of China, and the eastern English Channel. In particular, the northern Gulf of Mexico represents one of the largest areas of man-induced bottom-water anoxia in the coastal ocean, a so-called dead zone [15]. The most well-known blooms are related to the dinoflagellate *Karenia brevis* and diatoms of the genus *Pseudo-nitzschia* [14]. Brand and Compton in 2007 [16]

compared data from the periods between 1954–1963 and 1994–2002 and suggested that *Karenia brevis* was more abundant nearshore (12 to 18 times higher population abundance) due to the nutrient enrichment of nearshore coastal waters from human activities, such as urbanization and agriculture [16,17]. In the Chesapeake Bay, HAB events are related to the dinoflagellates *Karlodinium veneficum* (with blooms that have increased significantly and relate to fish kills [18,19]), *Prorocentrum minimum* (which forms more frequently high-biomass blooms), *Pfiesteria piscicida*, *Pfiesteria shumwayae*, and the pelagophyte *Aureococcus anophagefferens*. During the last decades, these events have increased in frequency and density in parallel with the increase of population, animal husbandry, and the usage of synthetic fertilizers [20]. HABs on the Chinese coast have increased in recent years in regards to their geographic extent, duration, and frequency, while a turnover in the causative algal species is observed [21]. From 2001 to 2010, dinoflagellates (e.g., *Noctiluca scintillans*, *Prorocentrum* sp., *Karenia mikimotoi*) and diatoms (e.g., *Skeletonema costatum*, *Pseudo-nitzschia pungens*) dominated, while the following years from 2011 to 2017, the haptophyte *Phaeocystis globosa* appeared as the dominant bloom-causing algal species [22]. From historical and empirical data, it can be extrapolated that industrial development, agriculture, and human activities were the main causes for the expansion of these blooms along the Chinese coast [23]. Finally, the eastern English Channel is affected by seasonal blooms of the haptophyte *Phaeocystis globosa*, a noxious rather than toxic species, due to its ability to produce foam from the lysis and degradation of its cells [24]. This foam negatively impacts fishery farming and bathing beaches due to conditions of low oxygen and increased viscosity [25,26]. The *Phaeocystis* blooms are suspected to be controlled mainly by nutrients originated from mainly anthropogenic input [27].

HABs can have adverse effects on human health, ecosystems, and human well-being through their impact on fisheries, tourism, and recreation [28]. The marine environment provides a great variety of benefits that are essential for our well-being. These benefits include the supply of food sources and biomedicines, as well as economic, psychological, and educational benefits. The occurrence of HABs decreases these benefits for people since they have negative impacts on human uses of marine ecosystem services. There is a growing interest concerning the importance of HABs and HAB-related illnesses to public health. HAB-related illnesses result from the consumption of contaminated seafood or through exposure to aerosolized toxins. The five most well-known HAB-related syndromes following seafood consumption are Ciguatera Fish Poisoning (CFP), Paralytic Shellfish Poisoning (PSP), Neurotoxic Shellfish Poisoning (NSP), Amnesic Shellfish Poisoning (ASP), and Diarrheic Shellfish Poisoning (DSP). In addition to that, other syndromes exist, such as Azaspiracid Poisoning (AZP), palytoxin poisoning, and tetrodotoxin poisoning [29,30]. These seafood-related poisonings cause issues with the following human systems, gastrointestinal, cardiovascular, neurological, and more rarely, coma and death. The effects arise few hours after consumption of fish and seafood contaminated with algae toxins. Recent reports show that aerosolization of some of these causative toxins from seawater could induce eye irritation and respiratory problems through inhalation [31]. Apart from these HAB-related syndromes, a growing interest has been developed among scientists concerning the effects of palytoxin-like toxins or "*Ostreopsis* spp. algal syndrome" in some Mediterranean beaches. This syndrome occurs with respiratory symptoms, skin irritation, and general malaise after exposure to seawater and/or aerosols during *Ostreopsis* spp. blooms [29]. Furthermore, HAB events characterized by high biomass accumulation could lead to environmental damage like hypoxia, anoxia, and decreased penetration of sunlight to submerged vegetation [32]. Mucilage or/and foam formation during HAB events through dissolved oxygen depletion and reduction of light penetration leads to severe damage to benthic organisms, negatively affecting the whole marine ecosystem [33]. Some toxins produced during HAB events are so strong that they can cause direct poisoning of fish and shellfish through neurotoxic or hemolytic effects, e.g., the dinoflagellate *Karenia brevis* produces toxins that paralyze the nervous system of fish, resulting in asphyxiation [11]. Finally, HABs can have negative economic and societal impacts, which are extremely im-

portant for local and regional economies and human well-being. Economic losses include costs from damages to aquaculture facilities due to lower production caused by illness and/or closure of monitoring programs, damages to seafood-dependent companies, and local service-based businesses due to reduced seafood consumption, and damages to the tourism industry (e.g., the presence of mucilage aggregates, foam, or water discoloration in coastal touristic areas provokes avoidance of these areas by visitors). Indirect economic impacts are also related to increased costs coming from more intense monitoring programs and increased health care services. Social impacts of HAB events arise from the loss of the aesthetic and recreational value of the sea, which disrupts the interaction between people and the marine environment and, in general, their connection with nature.

The aim of this review is to discuss the recent situation regarding the appearance of HABs in Mediterranean coastal areas associated directly or indirectly with anthropogenic eutrophication. First, HAB events reported in the Mediterranean Sea (MS) are presented to highlight the features and particularities of the MS and to summarize the harmful phytoplankton outbreaks along the length of coastal areas of many localities, i.e., Croatia, France, Italy, Spain, and Turkey. Second, we focus on HABs documented in Greek coastal areas (namely in the Gulfs of Evoikos, Malliakos, Amvrakikos, Kavala, Pagassitikos, Saronikos, and Thermaikos), according to the causative algal species, the period of occurrence and the induced damage in human and ecosystem health. The occurrence of HAB incidents during the past two decades is emphasized.

## 2. Literature Search Strategies

The databases of Elsevier Scopus (https://www.scopus.com/search/form.uri?display=basic#basic), Science Direct (https://www.sciencedirect.com/) and Google Scholar (https://scholar.google.gr/) were intensely searched (last accessed on 31 May 2021) by implementing the following keywords [eutrophication, coastal areas, urban areas, harmful algal blooms, marine phytoplankton, algae, toxins, Mediterranean sea], alone or in various combinations, in the fields "Article title, abstract, keywords" of Scopus and in the field "Keywords" of Science Direct, without any restraint of the report form. The search was performed initially by using a date range from 1960 to 2020; this produced thousands of documents. Particular focus was given on the results of the last two decades. The output data have been classified and analyzed, ruling out publications that did not fit the study's criteria. The publications regarding eutrophication and related harmful blooms in the coastal areas of MS show an increase in terms of the number of research articles in the last twenty years (Figure 1), which coincides with the development of advanced monitoring techniques of eutrophication and HAB dynamics. However, the decline in the number of research articles published in certain years can be related to the global economic crisis [34].

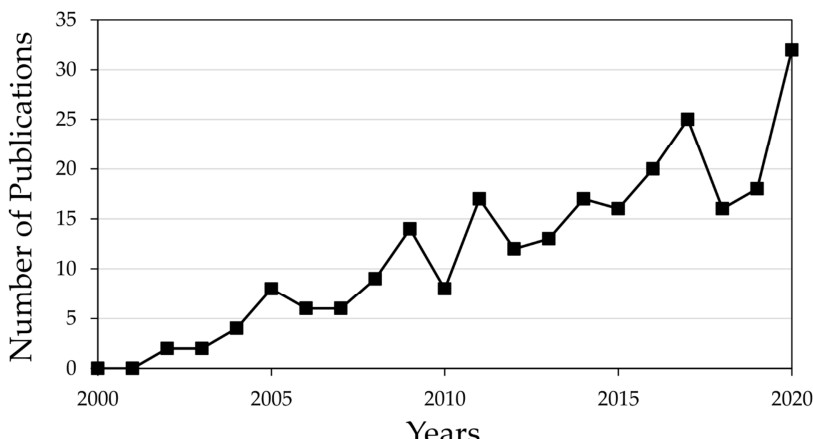

**Figure 1.** Number of published research articles on Harmful Algal Blooms (HABs) in the Mediterranean coastal areas during the period 2002–2020. No research articles were retrieved for the years 2000 and 2001.

### 3. HABs in the Mediterranean Sea (MS) Coastal Areas

The Mediterranean Sea (MS) is a large, enclosed sea that lies between Europe, Asia, and Africa. The MS is linked with the Atlantic Ocean through the Strait of Gibraltar in the west, is connected to the Red Sea by the Suez Canal in the southeast, and is also linked with the Black Sea in the northeast through the Strait of Gallipoli-Bosporus [35]. It covers an area of 25,000,000 km$^2$, equivalent to 1% of the Earth's surface, and includes ten different rather isolated sub-basins, which are: (1) Alboran Sea, (2) northwestern basin, (3) southwestern basin, (4) Tyrrhenian Sea, (5) Adriatic Sea (northern, central and southern), (6) Ionian Sea, (7) Central basin, (8) Aegean Sea (northern and southern), (9) North Levant Sea, and (10) south Levant Sea [36–38]. It represents a unique marine environment regarding its geomorphological, climatic, hydrological conditions, and biodiversity status. It is generally characterized by limiting nutrient conditions (especially due to low Phosphorus concentrations) which lead to low phytoplankton biomass and primary productivity. This oligotrophic complexion of MS increases from west to east and from north to south [39]. However, the coastal areas of MS are inhabited by many people with a coastal population density of 58.5 inhabitants per km$^2$ land area; the human population in these areas is expected to grow further in the following years and is projected to reach 200,000,000 inhabitants in 2025–2030 [40–42]. Thus, coastal areas of the MS are under numerous environmental pressures arising mainly from human activities. Drivers of marine ecosystem destruction and degradation include agriculture, aquaculture, animal farming, tourism and other economic activities, and discharge of untreated urban and industrial wastewaters. Following the increase of population in Mediterranean countries, anthropogenic activities increase the supply of nutrient inputs, which puts coastal zones under the threat of eutrophication [43,44]. According to a study based on the Geospatial Regression Equation for European Nutrient losses model (GREEN), during 2003–2007, 1.87 t year$^{-1}$ of total nitrogen (TN), and 0.11 t year$^{-1}$ of total phosphorus (TP) were discharged in the MS, whereas the Nile, Po, Rhone, Ebro, and northern Greece's rivers were the main suppliers of nutrient discharge. Furthermore, the study showed that the main contributor of TN is agriculture and that of TP is wastewater [43]. The nutrient-rich coastal areas represent an advantageous environment in which many algal species with harmful impacts might prevail [45]. Eutrophication of coastal areas mainly appears in enclosed bays, e.g., the northern Adriatic Sea, the Gulf of Lion, and the northern Aegean, and results in the appearance of HABs [30]. Moreover, climate change has a great effect on coastal areas as it alters some hydrological features of these areas, which further facilitates the occurrence of HABs [46]. HABs in coastal areas of the MS are various, extremely localized, and either appear seasonally every year or occur in an apparently stochastic manner. They may last from two weeks to two months depending on many components, which, separately or jointly, have influence over algal bloom dynamics [8]. The main accountable algal taxa for HABs events are dinoflagellates, *Alexandrium* spp., *Dinophysis* spp, *Ostreopsis* spp., and the diatoms *Pseudo-nitzschia* spp. [47]. Undoubtedly, among the coastal areas of Mediterranean localities, there are some which present a long history of HAB events and constitute a more representative example of anthropogenic eutrophication and related HABs. HABs documented in these regions are thoroughly presented below (Figure 2 for recorded bloom maxima in terms of number of harmful algal cells per region).

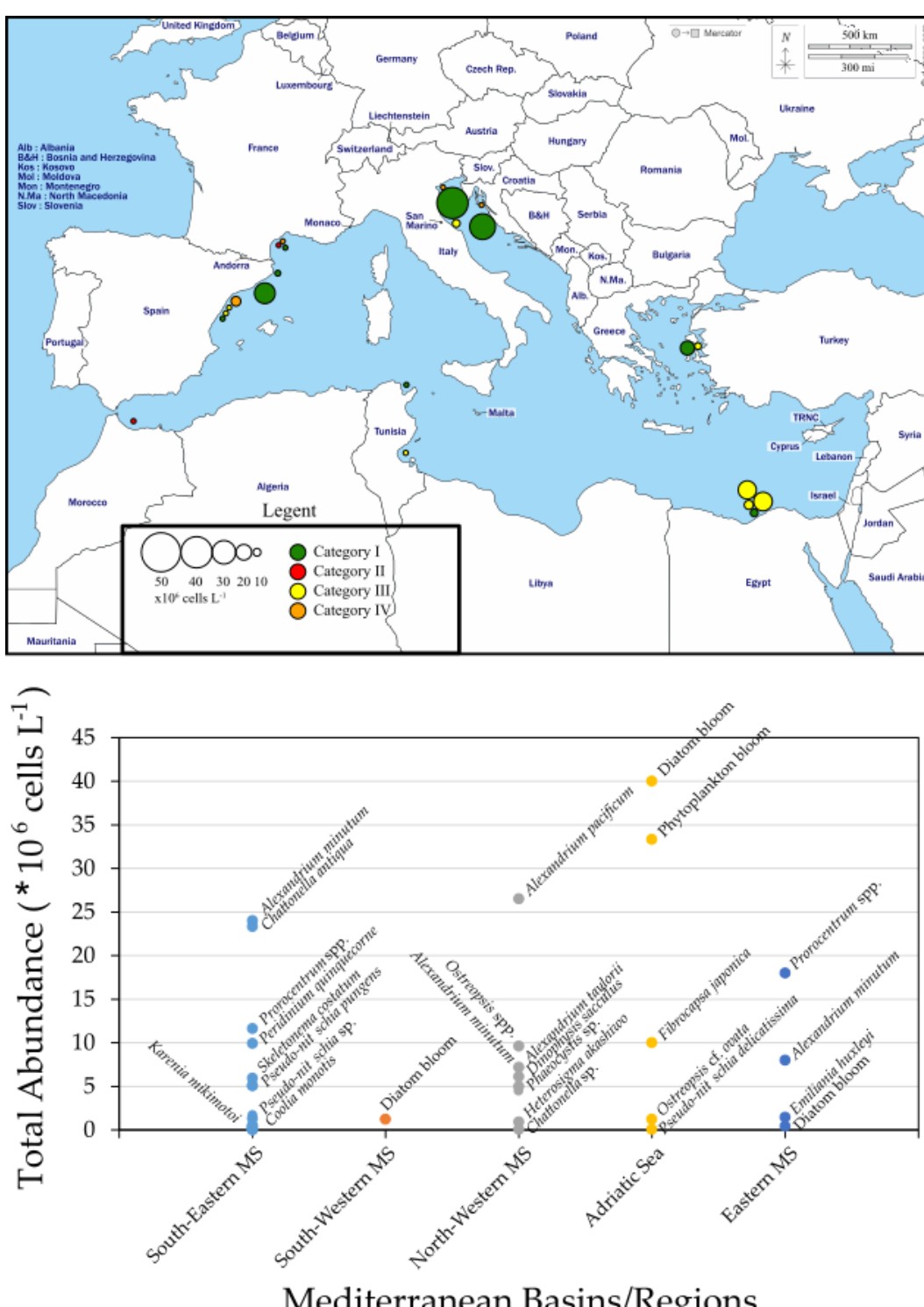

**Figure 2. Top**: major incidences of HABs in the MS. The size of the cycle represents the total abundance of the main taxa responsible for the blooms, and the color reflects the main category assigned to the HABs (see Table 1 for description of the four categories). **Bottom**: total abundance of phytoplankton in documented coastal HABs in the MS.

### 3.1. South-Eastern MS

HABs in the southeastern MS have been mainly reported from the coasts of Egypt and Tunisia. The Egyptian Mediterranean coast represents a "eutrophication hotspot" due to the intensive human pressure caused by anthropogenic nutrients which enter the sea

through coastal lagoons and land-based runoffs from the Nile. The annual total input of nutrients has been calculated at 676.4 t year$^{-1}$ for dissolved N and 84.9 t year$^{-1}$ for dissolved P [48]. The most eutrophic areas across the Egyptian coast are Abu Qir Bay, the eastern and western Harbors of Alexandria, and El-Mex Bay, whereas the Alexandria coasts, subject to continuous engineering modifications, receive the greatest pressure.

These modifications initially drove a rise in harmful algal species across the Egyptian Mediterranean coast from 29 to 38 species. However, subsequent corrective measures against this engineering alteration taken from the local authority in 2007 led to a decrease of the potentially harmful algal species from 38 to 17 [49]. The most well-known HAB event in the region appeared in the eastern harbor of Alexandria and was related to the dinoflagellate *Alexandrium minutum*, first reported in October of 1994, with a maximum abundance of $24 \times 10^6$ cells L$^{-1}$, resulting in mass fish/invertebrate kills [50]. Except for *Alexandrium minutum*, other dinoflagellate-triggered HAB events in the above-mentioned coastal areas include *A. monilatum* (maximum abundance of $1.3 \times 10^4$ cells L$^{-1}$ in June of 2006), *A. ostenfeldii* ($0.52 \times 10^6$ cells L$^{-1}$ in July of 2007, causing water discoloration and fish mortalities), and *Peridinium quinquecorne* ($9.9 \times 10^6$ cells L$^{-1}$ from August to September of 2007 resulting in water discoloration which had a negative impact on internal tourism). These harmful effects were strongly linked to coastal engineering modifications (TN: 3.43–22.37 µmol L$^{-1}$ and TP: 1.66–2.89 µmol L$^{-1}$) and were reduced from corrective measures taken after 2010. Other HABs included the proliferation of the dinoflagellates *Scrippsiella trochoidea* ($2.6 \times 10^6$ cells L$^{-1}$ in June 2006 resulting in water discoloration and mass kills of benthic organisms), *Gymnodinium* spp. ($1.17 \times 10^6$ cells L$^{-1}$ in August of 2005 resulting in invertebrate and fish kills), *Prorocentrum* spp. ($11.6 \times 10^6$ cells L$^{-1}$), *Ostreopsis* spp., *Karenia mikimotoi* ($1.32 \times 10^6$ cells L$^{-1}$ in August of 2007 resulting in fish kills), and *Coolia monotis* (blooms occurred during autumn of 2005–2006 and during the summers of 2007 and 2008 with a maximum abundance of $1.52 \times 10^4$ cells L$^{-1}$ in the summer of 2010). Moreover, the diatoms *Pseudo-nitzschia* sp. ($1.7 \times 10^6$ cells L$^{-1}$ in March of 2008 accompanied by high phytoplankton concentration, which had negative economic effects on tourism), *P. pungens* ($5.6 \times 10^4$ cells L$^{-1}$), and *Skeletonema costatum-complex* ($6.47 \times 10^4$ cells L$^{-1}$ in the winter of 2009), have also been causative algal agents for HAB phenomena in past years [51]. Additionally, the raphidophycean *Chattonella antiqua* caused massive monospecific blooms from late August to early September of 2006 and 2010 ($23.3 \times 10^6$ cells L$^{-1}$ in 2006, TN and TP concentration were calculated at 1.8 µmol L$^{-1}$ and 5.56 µmol L$^{-1}$, respectively) in Alexandria coastal waters resulting in mass fish kills [52].

The developing human activities (aquaculture, industry, tourism, maritime traffic, urbanization, etc.) lead to eutrophication impacts on the coastal areas of Tunisia. This cultural eutrophication resulted in the occurrence of several HABs. The most representative eutrophic coastal areas of Tunisia are the Gulf of Gabes and the Gulf of Tunis. The Gulf of Gabes has been recognized by general agreement among the eleven ecoregions of the Mediterranean Sea since it is characterized by the occurrence of algal blooms caused by nutrient inputs from industrial waste [53]. Heavy industrialization (phosphate production plants) of the nearby towns (Gabes, Sfax, Skhira) caused a great discharge of phosphogypsum which in turn has led to the marine ecosystem's degradation. Among the recorded algal species that cause HABs, the toxic dinoflagellates *Alexandrium minutum*, *Coolia monotis*, *Karenia selliformis*, *K. mikimotoi*, *Protoceratium reticulatum*, and *Ostreopsis* cf. *ovata* must be highlighted [54,55]. Dinoflagellates have been dominant in the algal community of the Gulf (recorded dinoflagellate maximum abundance $4.01 \times 10^5$ cells L$^{-1}$). Specifically, *Alexandrium minutum* has caused several blooms since 1990, resulting in water discoloration and fish mortalities. *Karenia selliformis* (which is responsible for the production of the toxins gymnodimines that contaminate the shellfish) and *Coolia monotis* were identified as the causative agents in harmful algae events, with extremely high concentrations, e.g., *Karenia selliformis* reached $2.5 \times 10^5$ cells L$^{-1}$ in October of 2009 accompanied by toxin production, and *Coolia monotis* peaked to $2.54 \times 10^6$ cells L$^{-1}$ in December of 2008 [56–58]. The Gulf of Tunis represents an urban and industrial area in Tunisia, and so it receives a large

amount of discharge from the lake of Tunis via the Rades channel [59,60]. Harmful algal blooms appeared during the previous years, triggered by dinoflagellates (dinoflagellate maximum abundance $4.8 \times 10^5$ cells $L^{-1}$) like *Dinophysis sacculus*, *Ostreopsis* cf. *siamensis*, and *Peridinium quinquecorne* [61,62]. Regarding diatom blooms, a maximum abundance of $3.85 \times 10^5$ cells $L^{-1}$ in the coastal waters of Tunisia is reported, with several species of *Pseudo-nitzschia* spp., *Nitzschia bizertensis*, and *Chaetoceros* spp. Although, there were no reports of impact on human health, probably due to the low levels of the produced toxins accumulated in bivalves [58,63,64].

### 3.2. South-Western MS

Mediterranean Moroccan coastal areas are exposed to several environmental disturbances originating from agriculture (phytosanitary treatments with chlorinated carbon compounds), heavy urbanization, aquaculture, and industrial activities. These urban and industrial discharges (over 650 industrial units along the Moroccan coastline) resulted in major hydrocarbon pollution of the sea, with TN concentrations reaching 7.52 μmol $L^{-1}$ and TP at 2.1 μmol $L^{-1}$) [65–67].

These human activities led to the loss of stability in phytoplankton communities and subsequently to the occurrence of harmful algal blooms. HAB incidences along the Mediterranean Moroccan coastline are mainly triggered by dinoflagellates and diatoms. The total abundance of phytoplankton has reached $1.2 \times 10^6$ cells $L^{-1}$, with diatoms being the most abundant in spring and summer [67]. Dinoflagellates like *Dinophysis* spp., *Prorocentrum lima*, *Alexandrium* spp., and *Gymnodinium catenatum* have been identified within Mediterranean Moroccan HABs. Regarding *Gymnodinium catenatum*, it is worth pointing out the severe and unique bivalve consumer's intoxication in 1994 (four deaths and twenty-three hospitalized people), which was attributed to Paralytic Shellfish Poisoning (*PSP*) according to mouse bioassays [40,68]. In recorded diatom blooms, the dominant algal species belong to *Pseudo-nitzschia* spp. These blooms have been frequently recorded since 2002 during the warm months, reaching the highest densities in May and September. On some occasions, the presence of domoic acid (DA) was detected in shellfish meat without exceeding the normative threshold [69].

### 3.3. North-Western MS

The Mediterranean Spain coastal zone is rather affected by the phenomenon of eutrophication mainly due to anthropogenic pressures, like agriculture (e.g., in Ebro Delta, rice field cultivation covers up to 65% of the area resulting in outputs of inorganic nutrients to nearby bays through drainage channels), aquaculture, tourism, construction of harbors (e.g., in Catalan coast, harbors increased from 12 in 1950 to 46 in 2003), intense urbanization, and industrialization [70–72].

There are records of several HAB incidences during the past years in the area, whereas the causative organisms are mainly dinoflagellates and diatoms but also haptophytes and raphidophytes. Many different dinoflagellates have been linked with HAB occurrences along the Mediterranean Spanish coastline, including species within the *Alexandrium* complex, belonging to *A. catenella* (which, in the MS, is most likely *A. pacificum*), *A. minutum*, *A. tamarense*, *A. taylorii*, *Karlodinium armiger*, *K. veneficum*, *Gymnodinium catenatum*, *Ostreopsis* cf. *ovata*, *O.* cf. *siamensis*, and *Dinophysis* spp. (*D. sacculus* mainly, but also *D. caudata*, *D. acuminata*). Especially for *Alexandrium pacificum* ($26.5 \times 10^6$ cells $L^{-1}$ in August of 2003) and *Alexandrium minutum* ($7.2 \times 10^6$ cells $L^{-1}$ from January to June of 2000), the Catalan coast represents an area of the world where heavy blooms of these species appear, causing water discoloration [73–76]. Furthermore, *Alexandrium taylorii* ($6.1 \times 10^6$ cells $L^{-1}$ from June to September of 2000) was firstly identified in 1997 at La Fosca beach of the Catalan coast, whereas the following years triggered recurrent blooms characterized by high biomass and discoloration of waters [77,78]. Regarding *Dinophysis* spp. (*D. sacculus* $5.9 \times 10^4$ cells $L^{-1}$ in spring of 2011), recurrent blooms located in two coastal embayments in the Ebro River Delta caused the closure of mussels' farms due to the presence of biotoxins on this extremely

important bivalve farming area of Catalonia [79,80]. The harmful algae events caused by *Ostreopsis* spp. (mainly *O.* cf. *ovata*) along the Catalan coast caused human intoxication due to aerosolized toxins in 2004 [81]. In addition, a dense bloom of *Ostreopsis* spp. was recorded from June to October of 2011, reaching its maximum abundance of $9.6 \times 10^6$ cells $L^{-1}$ in early July. Diatom blooms also frequently occur along the Mediterranean Spain coastal zone and are mainly attributed to the genus *Pseudo-nitzschia* ($>10^6$ cells $L^{-1}$); e.g., *P. calliantha*, *P. delicatissima*, *P. pungens*, *P. multistriata*, and *P. fraudulenta* [82]. Moreover, haptophyte proliferation caused by *Phaeocystis* sp. ($\geq 5 \times 10^6$ cells $L^{-1}$ in March of 2006) led to foam blooms and raphidophyte blooms of *Chattonella* sp. ($2 \times 10^6$ cells $L^{-1}$ in June of 2000), resulting in water discoloration and *Heterosigma akashiwo* ($2.1 \times 10^6$ cells $L^{-1}$ in July of 2007) resulted in mass fish mortalities.

In Mediterranean France, the Gulf of Lion is one of the most well-known eutrophic Mediterranean coastal areas. It represents an enclosed bay with particular features located in the north-western Mediterranean Sea, extending from Catalonia to Toulon. The Gulf's eutrophic status stems from receiving a large amount of rural, urbanized, and industrialized discharges through the Rhone River, which is the most important source of water and organic compounds in the MS [83–85]. The mean concentration of nitrate (more than 80% of TN-input from the Rhone to the MS) has increased by about 50% during the 1980s and 1990s. A similar TP increase is probable at the same time [84]. The most representative groups of phytoplankton identified along the Mediterranean French coast are dinoflagellates and diatoms. As far as dinoflagellates are concerned, the main causative species for the occurrence of HABs involved *Dinophysis* spp., mainly *D. acuminata* and *D. sacculus* (maximum annual abundance of $< 10^5$ cells $L^{-1}$, present in all seasons resulting in Diarrheic Shellfish Poisoning episodes), *Alexandrium* spp. (*A. minutum* resulting in PSP episodes, *and A. pacificum*) mainly in Thau lagoon and Bassin de Thau, *Ostreopsis* spp. (*O.* cf. *ovata*, *O.* cf. *siamensis*), and *Prorocentrum minimum*, which, for the first time, was identified and characterized from the Gulf of Lion [86,87]. Regarding *Ostreopsis* spp., nine blooms (maximum abundance of $9 \times 10^5$ cells $L^{-1}$ in August of 2006 with associated respiratory symptoms in several people) have been recorded from 2006 to 2009, where five of them caused skin and respiratory disorders in swimmers [88,89]. Diatom blooms along the Mediterranean French coast are mainly attributed to *Pseudo-nitzschia* spp. (maximum abundance $> 10^6$ cells $L^{-1}$) including *P. calliantha*, *P. delicatissima*, *P. pseudodelicatissima*, and *P. pungens,* but also to the genera of *Chaetoceros* (maximum abundance $> 10^6$ cells $L^{-1}$), *Skeletonema* (maximum abundance $> 10^6$ cells $L^{-1}$), and *Leptocylindrus* (maximum abundance $> 10^6$ cells $L^{-1}$). These blooms are observed in this particular area throughout the year, often resulting in Amnesic shellfish poisoning (ASP) episodes [90].

*3.4. Adriatic Sea*

The Italian coastline belongs partially to the Adriatic Sea, Ligurian Sea, Tyrrhenian Sea, and Ionian Sea. It should be mentioned that the Adriatic Sea (principally northern Adriatic) is one of the most productive and eutrophic basins in the MS, whereas the northern Adriatic represents the shallowest part of the Mediterranean (maximum depth < 200 m). The eutrophic status of the Adriatic's northern and western coasts is highly related to discharge from the Po River, making the Adriatic Sea an area of intense phytoplankton bloom formations and of several mucilaginous organic matter proliferation [91].

The causative organisms for these large algal blooms in the northern Adriatic Sea are mainly dinoflagellates and diatoms. The last few years have exhibited a shift regarding their systematic seasonal pattern [92]. The total abundance of phytoplankton in bloom events has reached $40 \times 10^6$ cells $L^{-1}$ with a remarkable presence of diatoms [93]. During these bloom formations, from October to December of 2000, the nutrient input from Po River was 8969 and 650 tons for TN and TP, respectively. Regarding dinoflagellates, different species have been associated with HAB events, including *Dinophysis* spp., resulting in DSP episodes (*D. tripos*, *D. sacculus*, and *D. caudata*), *Alexandrium* spp., (mainly *A. minutum* related to PSP episodes, and *A. mediterraneum*, *A. pseudogonyaulax*, *A. tamutum*, *A. taylori*), *Gonyaulax* spp.,

*Noctiluca scintillans* ($>2 \times 10^3$ cells L$^{-1}$ resulting in the discoloration of waters in Gulf of Trieste during the winter of 2002–2003), *Prorocentrum* spp., and *Ostreopsis* cf. *ovata* [94–96]. From 2006, and in accordance with other coastal Mediterranean areas, severe blooms of *Ostreopsis* cf. *ovata* have occurred in the northern Adriatic Sea, resulting in aerosolized toxins with effects on humans and benthic organisms [97,98]. Diatoms occur rather frequently on Italian coasts leading to HAB incidences. The most representative species of diatoms are *Skeletonema marinoi* (which causes blooms seasonally in the northern Adriatic Sea, which have recently shifted from winter to spring), *Pseudo-nitzschia* spp., (*P. delicatissima*, *P. pseudodelicatissima*, *P. multistriata*), *Chaetoceros* spp., and *Cylindrotheca closterium*. It should be mentioned that diatoms, especially *Cylindrotheca closterium*, along with the dinoflagellate *Gonyaulax fragilis,* are the main triggering factors of mucilage formation episodes, "the dirty sea phenomenon", which have been conspicuous in the northern Adriatic Sea since the 1990s [99,100]. Besides diatoms and dinoflagellates, raphidophytes, like *Fibrocapsa japonica* (a potentially ichthyotoxic species), have caused recurrent blooms from the 1990s (maximum abundance of $10 \times 10^6$ cells L$^{-1}$ in 2004 and 2006), whereas, after 2012, these became less frequent [101].

The Croatian coastline, mainly in the northeastern part of the Adriatic Sea, is strongly influenced by the Po River's freshwater discharge, which has a major impact on nutrient inputs and phytoplankton abundance. The most frequent and abundant taxa of phytoplankton in the area comprise diatoms and dinoflagellates, whereas the total abundance of phytoplankton in bloom incidences reached $33.3 \times 10^6$ cells L$^{-1}$. Moreover, these blooms during the following years showed a shift in their seasonality. Diatoms represent the most dominant group in most areas of the northern Adriatic Sea throughout the whole year, while dinoflagellates mostly dominate in areas directly influenced by river discharges [102]. Concerning dinoflagellates, some of the species involved are *Dinophysis* spp. (*D. tripos*, *D. sacculus*, *D. caudata*, *D. fortii*), *Alexandrium* spp., *Lingulodinium polyedra*, *Gonyaulax spinifera*, *Prorocentrum* spp., and *Ostreopsis* cf. *ovata*. Especially for *Ostreopsis* cf. *ovata* a large bloom incidence occurred on a public beach near Rovinj (Croatian coast, N. Adriatic Sea) that caused respiratory problems to humans that visited the beach. During this bloom, lasting from September to October of 2010, the dominant *Ostreopsis* cf. *ovata* peaked at $4.26 \times 10^4$ cells L$^{-1}$ [103]. Diatom blooms are mainly attributed to *Pseudo-nitzschia* spp., mostly the *P. delicatissima* group, which is present in coastal waters during the whole year, reaching densities up to $1.2 \times 10^6$ cells L$^{-1}$, and resulting in the accumulation of DA in shellfish without exceeding the normative threshold. Other harmful diatoms have also been detected, such as *Chaetoceros* spp., *Cerataulina pelagica*, and *Cylindrotheca closterium* [104–106].

### 3.5. Eastern MS (Aegean Sea)

The Aegean Sea is part of the eastern Mediterranean basin. A part of the Aegean coastline belongs to Turkey, while another part belongs to Greece [107]. The Aegean coastal areas have been subject to many pressures like environmental degradation (increased population density, increased wastewater discharges, agriculture, aquaculture, industrialization, tourism, maritime transportation) and climate change. All these stresses force the marine ecosystem to respond via an increase in the frequency and intensity of HABs, which characterize the Aegean nearshore coastal areas. More specifically, there are some coastal areas of Turkey that are considered hotspots of eutrophication and related HABs, like Izmir Bay, a large embayment located in the eastern coast of the Aegean Sea, which is heavily polluted by nutrients and organic materials due to domestic and industrial discharge [108,109].

Several HAB incidences have been reported during the past years along the Turkish Aegean coastline, whereas the causative organisms are mainly dinoflagellates and diatoms but also haptophytes. The total abundance of phytoplankton in some bloom events exceeded $10^6$ cells L$^{-1}$. Regarding dinoflagellates, they dominate dense blooms in this area, whereas there are records of bloom incidences mostly in Izmir Bay, Gulluk

Bay, and also Canakkale Strait. One of the species involved was *Alexandrium minutum*, which caused a red tide event with mass fish kills in 1983 and 1998 at Izmir Bay with cell abundances that reached $8 \times 10^6$ cells $L^{-1}$ and $4.1 \times 10^5$ cells $L^{-1}$, respectively [110]. The most noticeable and widespread blooms were caused by *Prorocentrum* spp., mainly *P. micans*. A high biomass-forming dinoflagellate-causing red water discoloration with abundances reaching $18 \times 10^6$ cells $L^{-1}$ occurred in May of 2015, concurrent with elevated levels of $NH_3$ (0.28 mg $L^{-1}$) and TP (42.0 mg $L^{-1}$), but also *P. minimum*, *P. scutellum*, *P. lima*, *P. triestinum*, and *Noctiluca scintillans*, which represent the most frequent causative species for dense red tides causing pale red water discoloration with abundances reaching up to $> \times 10^5$ cells $L^{-1}$ [108,111,112]. Additionally, other dinoflagellates have been identified in bloom events as *Gymnodinium catenatum*, *Karenia mikimotoi*, *Dinophysis* spp., *Gonyaulax fragilis* that causes mucilage formation, *Ceratium* spp., and *Scrippsiella trochoidea*. Diatom blooms with abundance $> 10^5$ cells $L^{-1}$ are mainly attributed to *Skeletonema costatum-complex*, *Thallassiosira* sp. *Cylindrotecha closterium*, and *Pseudo-nitzschia pungens* [113]. Furthermore, regarding mucilage events that were first documented in 2007 in Izmir Bay, the most abundant species involved were *Skeletonema costatum-complex*, *Cylindrotecha closterium*, *Thallassiosira rotula*, and *Gonyaulax fragilis*. Apart from diatoms and dinoflagellates, haptophytes like *Emiliania huxleyi* caused tides with a "milky sea" appearance, with abundances peaking at $1.44 \times 10^6$ cells $L^{-1}$ [114].

## 4. HABs in Coastal Areas of Greece

The Greek coastline, which is about 18,000 km in length, is part of the eastern MS, is encircled by the Aegean, Ionian, and Cretan Seas, and has a rather diverse ribbon-like morphology. This morphology is characterized by the presence of various enclosed and semi-enclosed gulfs, which often show signs of eutrophication, mainly anthropogenic, due to nutrient enrichment from rivers and other water outfalls stemmed from agricultural, aquacultural, urban and industrial activities. A growing interest in harmful algae and related blooms in Greek coastal areas has developed since 1977 when a red-brown phytoplankton bloom of *Karenia brevis*-like species with abundance $10 \times 10^6$ cells $L^{-1}$ was recorded in Saronikos Gulf coinciding with mass fish kills and was attributed to the increased anthropogenic effects [115]. The most frequently detected HAB taxa in coastal waters of Greece include diatoms, dinoflagellates, haptophytes, and raphidophytes. Some of these species, such as *Pseudo-nitzschia* spp., *Dinophysis acuminata*, and *Karenia brevis* could be potentially toxic, and others like *Alexandrium insuetum* and *Noctiluca scintillans* are high-biomass forming [116]. The Gulfs of Saronikos, Evoikos, Pagassitikos, Amvrakikos, Thermaikos, and Kavala are the main areas where the occurrence of HABs have been documented [36,117]. HABs in Greek coastal areas are presented below (Figure 3 for recorded bloom maxima in terms of number of harmful algal cells per Greek coast).

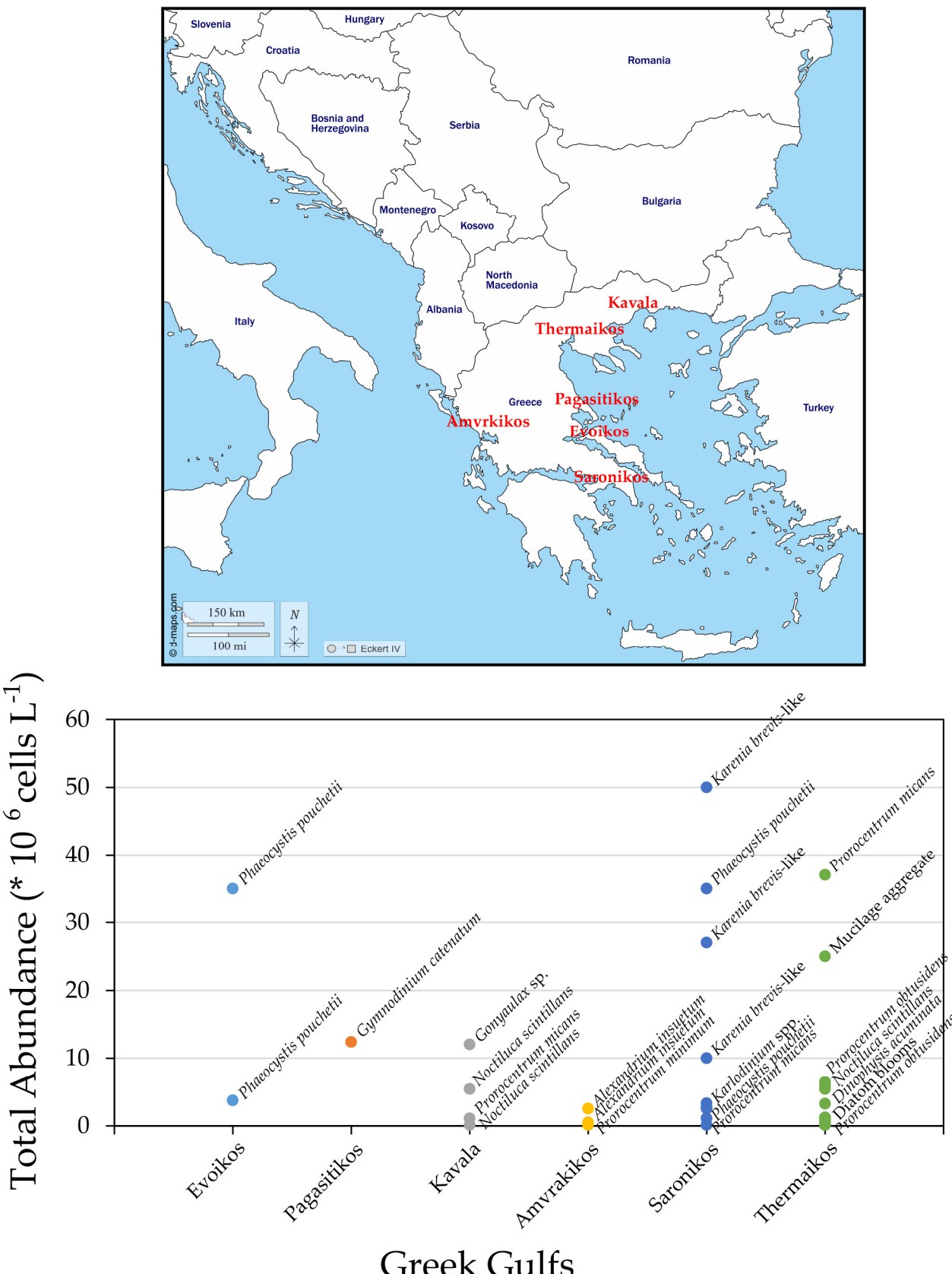

**Figure 3. Top**: Map of the Baltics with focus on the Greek coastal areas that HABs have been documented. **Bottom**: Total abundance of phytoplankton in documented HABs in Greek Gulfs.

### 4.1. Saronikos Gulf

Saronikos Gulf is one of the largest semi-enclosed embayments of the central Aegean Sea. The Gulf contains many islands and is divided into the eastern and western parts. The northeastern part of the Gulf (inner Gulf) is rather shallow with a mean depth of 90 m, while the western part has a maximum depth of 450 m to the south [118,119]. The Saronikos and Thermaikos Gulfs are the most strongly affected Greek coastal areas by anthropogenic eutrophication. The main human activities which influence the coastal marine ecosystem of the Saronikos Gulf are agricultural and tourist activities, pollution from nearby industrial units, extensive engineering modifications of the coastline, industrial and commercial shipping and shipbuilding activities, heavy urbanization of Athens and Piraeus, and urban sewage discharge [120].

In September 1977, September 1978, and October 1987, massive HAB events due to *Karenia brevis*-like species were reported ($10 \times 10^6$ cells $L^{-1}$, $50 \times 10^6$ cells $L^{-1}$, and $27 \times 10^6$ cells $L^{-1}$, respectively), resulting in mass fish kills. In March 1989, August 1993, and September 1999, *Phaeocystis pouchetii* ($2.5 \times 10^6$ cells $L^{-1}$, $35 \times 10^6$ cells $L^{-1}$, and $2.7 \times 10^6$ cells $L^{-1}$, respectively) caused water discoloration, mucilage formation, and degradation of coastal areas. In April 2003, *Prorocentrum minimum* ($1.1 \times 10^5$ cells $L^{-1}$) caused water discoloration and visual nuisance for the locals. In May 2003, *Prorocentrum micans* ($1.1 \times 10^6$ cells $L^{-1}$) also had the same effects [116,121]. In December 2014, a HAB event (due to the ichthyotoxic *Karlodinium* spp.) occurred in Saronikos (Salamina island) with a maximum abundance of $3.3 \times 10^6$ cells $L^{-1}$, causing water discoloration and resulting in mass fish kills and a visual nuisance for the local people. In April 2016, a HAB event due to *Noctiluca scintillans* caused water discoloration and visual nuisance for the locals.

### 4.2. Thermaikos Gulf

Thermaikos Gulf is a semi-closed embayment of the northwest Aegean Sea with a maximum depth of 45 m. The Gulf is divided into the inner and outer parts, whereas the inner part includes the Thessaloniki Bay and the commercial harbor of Thessaloniki and is regarded as one of the most human affected coastal areas of Greece and, more generally, of the eastern Mediterranean [122]. The eutrophic status of the Gulf is strongly related to the nutrient inputs originating from river discharges and human activities. As for river discharges, the Gulf receives great amounts of water from the Axios, Aliakmon, Loudias, and Gallikos Rivers, with the Axios River having the highest contribution. Human pressures derive from many activities like agriculture on the coastal plains, aquaculture (especially the shellfish and mainly the mussels' production represents about 80–85% of the national production, whereas the mussels' rafts through the change of the microhabitats favor the growth of dinoflagellates), urban and industrial wastewaters, construction of dams, tourism, and shipping [123–125].

In the Thermaikos Gulf, recurrent HAB events have been reported during the past years, characterized by water discoloration, mucilage aggregation, or toxic events. Furthermore, the majority of HAB events associated with water discoloration in Greece were recorded in the Thermaikos Gulf [126]. In the spring of 1994 and winter of 2000 and 2001, species of the genus *Prorocentrum* caused HAB events characterized by water discoloration, a visual nuisance for local people, and possible local anoxia episodes. Specifically, blooms of *P. micans* (April of 1994, $37 \times 10^6$ cells $L^{-1}$), *P. obtusidens* (January of 2000 and 2001, $1.2 \times 10^6$ cells $L^{-1}$), and *P. redfeldii* (winter of 2000 and 2001, $1.2 \times 10^6$ cells $L^{-1}$ and $6 \times 10^6$ cells $L^{-1}$, respectively) were identified. Bloom events caused by *Dinophysis acuminata* are also frequently reported in the Thermaikos Gulf related to shellfish deaths and economic losses for the shellfish industry. In chronological order, the first HAB event of *D. acuminata* was reported in 2000; the bloom began in January and lasted until the end of March, peaked at $8.5 \times 10^4$ cells $L^{-1}$, was the first toxic event reported in Greece, and was classified as such based on DSP symptoms on humans and mouse bioassays [127]. During this *D. acuminata* bloom, the algal community was also dominated by the diatoms

*Skeletonema costatum-complex* and *Leptocylindrus minimus*, with a maximum abundance of $3.26 \times 10^6$ cells L$^{-1}$ and $3.06 \times 10^6$ cells L$^{-1}$, respectively. In 2002, the *D. acuminata* bloom began in January and lasted until the end of April, peaking at $3.7 \times 10^4$ cells L$^{-1}$ in February. During this bloom, the algal community was also dominated by the diatoms of the genus *Pseudo-nitzschia*, with a maximum abundance of $7.83 \times 10^5$ cells L$^{-1}$. More bloom events caused by *D. acuminata* were also documented in March 2003 ($2.2 \times 10^3$ cells L$^{-1}$) and in May 2004 ($1.1 \times 10^4$ cells L$^{-1}$) [116,128–130]. Another dinoflagellate, the species *Noctiluca scintillans*, triggers recurrent HAB episodes mostly in late winter-early spring, which are linked to discoloration and high viscosity of the water. In particular, blooms of *N. scintillans* were reported in the last twenty years, reaching maximum abundances of $5.4 \times 10^6$ cells L$^{-1}$ in 2015, $3.25 \times 10^6$ cells L$^{-1}$ in 2017, and $2 \times 10^5$ cells L$^{-1}$ in January and November 2019. In some cases, like in 2017, *N. scintillans* co-occurred with the dinoflagellates *Spatulodinium pseudonoctiluca* and *Gonyaulax fragilis,* and other mucilage-producing and bloom-forming diatoms, such as *Cylindrotheca closterium*, *Chaetoceros* spp., *Leptocylindrus* spp., *Skeletonema costatum-complex*, and the haptophyte *Phaeocystis* sp. in high abundances reaching $25 \times 10^6$ cells L$^{-1}$. The co-existence of all these species led to an intense mucilage aggregate event in late June named a "dirty sea phenomenon" [131,132]. Furthermore, the dinoflagellate *Gonyaulax spinifera* in October-November of 2013 caused a HAB event lasting for three weeks reaching maximum abundances of $6.4 \times 10^5$ cells L$^{-1}$, with brownish water discoloration [133]. The ichthyotoxic dictyochophycean *Vicicitus globosus* triggered HAB events with maximum abundances $> 10^4$ cells L$^{-1}$ during the spring of 2001–2003, accompanied by water discoloration. Furthermore, the photosynthetic ciliate *Mesodinium rubrum* caused a red tide event from December 2017 to February 2018 with a maximum abundance $> 10^6$ cells L$^{-1}$ [132]. Finally, *Vicicitus globosus* in April of 2018 caused water discoloration resulting in visual nuisance for the locals.

*4.3. Other Greek Gulfs*

The Evoikos Gulf is a confined embayment extending along the coast of Evia, which is characterized by a tidal phenomenon. It is divided into north and south Evoikos and is affected by eutrophication due to urban and industrial discharges [134]. In August of 1993, a HAB event with mucilage formation was attributed to *Phaeocystis pouchetii* (maximum abundance $35 \times 10^6$ cells L$^{-1}$), resulting in the degradation of the coastal area. Moreover, in September of 1999, a severe algal bloom occurred in the area characterized by high phytoplankton concentrations and mucilage formation resulting in mucus-forming ''blankets'' floating on the water surface. Four phytoplankton taxa, including cyanobacteria, dinoflagellates, diatoms, and haptophyta, caused this bloom event. Regarding diatoms, the dominant species were *Climacosphenia moniligera*, *Navicula forcipata*, and *Nitschia closterium* with abundances of $94.6 \times 10^4$ cells L$^{-1}$, dinoflagellates were dominated by *Protoperidinium sphaeroides* (abundance $5.3 \times 10^4$ cells L$^{-1}$), and haptophytes were dominated by *Phaeocystis pouchetii* (abundance $2.7 \times 10^6$ cells L$^{-1}$) [135].

Pagasitikos Gulf is a semi-enclosed embayment with a maximum depth of 108 m. At the northern part of the Gulf, which encircles the coast of Volos bay, is located the industrial city and also the port of Volos. Eutrophication of the Gulf is strongly connected to nutrient pollution due to rural, industrial, and agricultural discharges from fertilized fields [136,137]. In July of 1987, at Volos harbor, a red-brown phytoplankton bloom appeared (maximum total phytoplankton abundance of $12.4 \times 10^6$ cells L$^{-1}$) in which the dominant species was the dinoflagellate *Gymnodinium catenatum* reaching a density of 89.7% of the total abundance [138]. Moreover, in April of 2017, a HAB event occurred characterized by water discoloration and high phytoplankton concentration resulting in visual nuisance for local people, whereas the causative species was identified as *Alexandrium minutum*.

Kavala Gulf is a semi-enclosed embayment in the north Aegean Sea with a maximum depth of 60 m. The Gulf is rather affected by eutrophication due to nutrient influents of the port, the city's sewage treatment operations, the nearby industries, and the Nestos river lagoons [139,140]. In March of 1978, a bloom event with water discoloration appeared near

the port of Kavala following the wreck of a ship loaded with phosphate fertilizers. The causative species was *Noctiluca scintillans* ($1.1 \times 10^5$ cells L$^{-1}$). In August of 1986, the proliferation of *Gonyaulax* sp. ($12 \times 10^6$ cells L$^{-1}$) caused a red-brown water discoloration [141]. In May of 1993, a HAB event caused by *Prorocentrum micans* ($1.1 \times 10^6$ cells L$^{-1}$) resulted in water discoloration and a visual nuisance for locals. Furthermore, during the years 2000–2004, from February to March, *Noctiluca scintillans* ($5.4 \times 10^6$ cells L$^{-1}$) frequently triggered HAB events resulting in water discoloration [115,116].

Maliakos Gulf is a coastal area located in central Greece, which receives Spercheios river discharges. The Gulf is divided into the inner (maximum depth 25 m) and outer parts (maximum depth 50 m). Except for river discharges, the Gulf is also under anthropogenic eutrophication, with common pressures being agriculture, aquaculture, urban and industrial sewage, and fishing [142,143]. In March-April of 2009, a bloom event caused by the raphidophyte *Chattonella* sp. resulted in massive fish kills with significant economic losses [144].

Amvrakikos Gulf is a semi-enclosed bay in northwestern Greece with a maximum depth of 65 m, whereas it is connected to the Ionian Sea by a narrow opening—the Preveza channel. The Gulf receives river discharge through a drainage basin from Arachthos and Louros rivers, resulting in nutrient inputs—mainly phosphates [145,146]. Other sources of nutrient pollution in the Gulf include agriculture, fish farming, and urban sewage [147]. In December of 1998, a HAB event stimulated by the massive presence of the raphidophyte *Pseudochattonella verruculosa* resulted in mass fish kills. The dinoflagellate *Prorocentrum minimum* occurs with high abundances every year during autumn in the area, whereas in September of 2003, it triggered a HAB event characterized by water discoloration ($10^5$ cells L$^{-1}$). In April 2003 and May 2004, *Alexandrium insuetum* reached maximum abundances of $2.5 \times 10^6$ cells L$^{-1}$ and $4.7 \times 10^5$ cells L$^{-1}$, respectively, causing a brownish water discoloration resulting in a visual nuisance for locals [126].

## 5. Conclusions and Perspectives

HABs constitute a major global problem due to their tendency to cause environmental damage, healthcare issues, and economic losses. This complex and diverse issue, driven by climate change and increased human activities in coastal areas, needs to be addressed so that, in the long run, the major negative impacts of HABs in the coastal areas of the Mediterranean Sea can be diminished. Preventive measures aim to control blooms before their development and are mainly focused on avoiding eutrophication representing the most critical component of most management strategies. These measures are centered on the decrease of nutrient inputs to the sea, including industrial, urban, and agricultural sewage. The implementation of policies according to the EU legal framework (Nitrates Directive 91/676/EEC [148] and Urban Waste Water Directive 91/271/EEC [149]) led to the reduction of agricultural nutrient inputs and wastewater discharges. This decrease has been accomplished in some Mediterranean areas, e.g., the north Adriatic Sea (Po river basin), northwestern MS (Rhone basin), and the Aegean Sea. However, eutrophication problems persist in the Nile Delta area [44]. Furthermore, Mediterranean countries (not Member States) designed their legislation in line with the EU legislation [36]. A significant role for the prevention and control of HABs includes monitoring programs for toxin and cell detection and the quantification in seawater and meat of fish/shellfish using laboratory-based methods. Aside from these monitoring programs, some countries also use other methods for the early detection of HABs, including satellite remote sensing and complex numerical models.

Management strategies constitute remedial measures after the outbreak of HABs to control and decrease their growth [11,32]. These management strategies could include physical, chemical, and biological approaches. Physical methods in coastal areas are related to aquaculture facilities, aiming towards the separation of fish and HABs, and providing air to prevent anoxia. Such methods include the use of perimeters skirts in combination with pump aeration in pens. Chemical methods involve the use of some algicides like copper

sulfate ($CuSO_4$). Until now, this is the only chemical control intervention in seawater that has been recorded. However, it should be stressed that the use of chemicals in sea areas is not a preferred method due to potential indirect negative impacts on the ecosystem. Physical-chemical methods refer mainly to the use of flocculating agents for inducing coagulation-flocculation and sedimentation of harmful microalgae. Such flocculating agents are mainly sand or clay, although the use of clays is considered the most effective measure for HABs control in the sea since they are not toxic and don't have important ecological impacts [150]. Biological methods include the use of biological agents acting as controllers of HABs through feeding (predators), infecting, or decomposing (parasites, bacteria, fungi, viruses) HAB species. Although biological control methods seem promising, more research is needed to fully implement them in the field due to their possible environmental effects.

Notwithstanding, there are many scientific advances regarding management systems; however, HAB prevention, which is extremely important, could be further enhanced. Some suggestions include:

- rectification of farming and animal husbandry methods according to good agricultural practices;
- establishment of proper and complete physical/chemical wastewater treatment or/and wastewater treatment in algal ponds which enable circular bio-economy [151];
- collecting and disposing of dead coastal organisms via burial or rendering;
- more frequent, systematic, and expanded monitoring programs for harmful algae and algal toxin detection;
- development and application of advanced technologies like the -omics technologies to better understand the diversity, evolution, ecology, and dynamics of HABs;
- outreach and education of public on HAB occurrence and impacts;
- establishment of a reporting system engaging the local community, such as the type that has been developed in the Gulf of Mexico, to provide daily updates from trained volunteers during algal blooms [31,152];
- co-operation between regulatory authorities, organizations, academic institutes, researchers, and citizens to exchange relevant knowledge and best practices.

**Author Contributions:** Both authors contributed equally to the writing of the paper. C.T. performed literature searches and wrote an initial draft. Both authors have read and agreed to the published version of the manuscript.

**Funding:** This research received no external funding.

**Conflicts of Interest:** The authors declare no conflict of interest.

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
