# Peer review of "Review of Harmful Algal Blooms in the Coastal Mediterranean Sea, with a Focus on Greek Waters"

_diversity, doi:10.3390/d13080396_

Round 1
Reviewer 1 Report
GENERAL COMMENTS
In this review, Authors gathered information from several papers about harmful algal blooms in Mediterranean Sea with particular focus on results of the last two decades. Although this review is not so original because of the existence of another review (recently published in Harmful Algae) about HAB in Mediterranean Sea (Zingone, A.; Escalera, L.; Aligizaki, K.; Fernandez-Tejedor, M.; Ismael, A.; Montresor, M.; Mozetič, P.; Taş, S.; Totti, C. Toxic marine microalgae and noxious blooms in the Mediterranean Sea: A contribution to the Global HAB Status Report. Harmful Algae 2021, 102, 101843), in this review additional data are shown and a particular focus was given to the Greek coastal areas. For this reason, in my opinion the paper deserves to be published after minor changes/improvements suggested in my comments.
SPECIFIC COMMENTS
- Table 1. this table is not exhaustive as several species treated in this review are not listed. For example, in line IV, Ostreopsis is missing. Please implement tis list with all the species you are talking about in your review.
- L78-80. In Chesapeake Bay, also Karlodinium veneficum should be mentioned, adding further and more recently references.
- Figure 2. To make this figure more readable, change 1, 2, 3, 4 and 5 to South-Eastern MS, South-Western MS, North-Western MS, Adriatic Sea and Eastern MS.
- Figure 3. To make this figure more readable, change 1, 2, 3, 4, 5 and 6 to Evoikos Gulf, Pagasitikos Gulf, Kavala Gulf, Amvrakikos Gulf, Saronikos Gulf and Thermaikos Gulf.
- Add a figure showing a map indicating the position of the Evoikos Gulf, the Pagasitikos Gulf, the Kavala Gulf, the Amvrakikos Gulf, the Saronikos Gulf and the Thermaikos Gulf.
- Impacts of HABs. This part seems to be inappropriate here. I suggest merging it with the conclusion section or move it in the introduction part.
Author Response
Reviewer 1
GENERAL COMMENTS
In this review, Authors gathered information from several papers about harmful algal blooms in Mediterranean Sea with particular focus on results of the last two decades. Although this review is not so original because of the existence of another review (recently published in Harmful Algae) about HAB in Mediterranean Sea (Zingone, A.; Escalera, L.; Aligizaki, K.; Fernandez-Tejedor, M.; Ismael, A.; Montresor, M.; Mozetič, P.; Taş, S.; Totti, C. Toxic marine microalgae and noxious blooms in the Mediterranean Sea: A contribution to the Global HAB Status Report. Harmful Algae 2021, 102, 101843), in this review additional data are shown and a particular focus was given to the Greek coastal areas. For this reason, in my opinion the paper deserves to be published after minor changes/improvements suggested in my comments.
We would like to thank the reviewer for his/her positive disposition towards our work. All comments and suggestions have been taken into consideration and the necessary changes have been highlighted in red in the revised manuscript to facilitate reading.
SPECIFIC COMMENTS
- Table 1. this table is not exhaustive as several species treated in this review are not listed. For example, in line IV, Ostreopsisis missing. Please implement tis list with all the species you are talking about in your review.
Thank you for the suggestion, we have now modified the table accordingly.
- L78-80. In Chesapeake Bay, also Karlodinium veneficumshould be mentioned, adding further and more recently references.
Thank you for the suggestion, we have now mentioned Karlodinium veneficumadding the relevant references.
- Figure 2. To make this figure more readable, change 1, 2, 3, 4 and 5 to South-Eastern MS, South-Western MS, North-Western MS, Adriatic Sea and Eastern MS.
Thank you for the suggestion, we have now modified the figure and the figure legend accordingly. We have also included the main HAB species involved in each case, and a map of the Mediterranean Sea with graphical representation of major bloom events, as another reviewer suggested. We hope that this meets your requirements.
- Figure 3. To make this figure more readable, change 1, 2, 3, 4, 5 and 6 to Evoikos Gulf, Pagasitikos Gulf, Kavala Gulf, Amvrakikos Gulf, Saronikos Gulf and Thermaikos Gulf.
Thank you for the suggestion, we have now modified the figure accordingly and the figure legend. We have also included the main HAB species involved in each case, and a map of the Aegean Sea including the Greek gulfs where major bloom incidents have appeared and are mentioned in-text and in the graph. We hope that this meets your requirements.
- Add a figure showing a map indicating the position of the Evoikos Gulf, the Pagasitikos Gulf, the Kavala Gulf, the Amvrakikos Gulf, the Saronikos Gulf and the Thermaikos Gulf.
Thank you for the suggestion, we have now modified Figure 3 to include the position of the Greek Gulfs.
- Impacts of HABs. This part seems to be inappropriate here. I suggest merging it with the conclusion section or move it in the introduction part.
We agree with the reviewer and believe that this part is more suited to the introduction. Thus, we have moved it in L84-114 of the introduction.
Reviewer 2 Report
The value of this work is that it presents a comprehensive literature review (146 references) of Harmful Algal Bloom events in Mediterranean coastal waters, with a focus on Greek waters. A limitation is that no effort is made to produce a quantitative analysis of trends (only Fig 1 on number of publications), nor distribution maps presented (this is essential to include for non-Mediterranean readers; ideally all regions mentioned in the text should be labelled on a map), nor the severity of economic impacts quantified. The extensive claims of links with eutrophication are not quantified (line 18 in abstract; no nutrient data are presented!!); As admitted on line 67, these linkages are not always clear; note that only high biomass blooms (category 1 in Table 1; line 53) are commonly related to nutrients, but many other HABs such as Alexandrium can cause toxic problems at extremely low cell concentrations. There exist significant overlap with the excellent 2021 Zingone et al. Harmful Algae review (cited in ref 32; 501 HAEDAT events covered; 600 publications; 2350 Mediterranean records of 84 HAB species)), and this papers should attempt to differentiate itself eg., by the suggested modified title "Literature review of HABs in Mediterranean Sea, with a focus on Greek coastal waters". [there is no recent history presented; no time series analysed).
Analyse the literature data in Fig.1 better; break down into HAB categories; events in the 5 regions here discriminated etc; whether papers relate HAB to eutrophication etc.,
line 30. ...marine environments, and can be expressions of ecosystem disturbance.
line 31. ..population live..
This is not a global review, therefore all examples in Table 1 should be confined to Mediterranean waters. Delete Pyrodinium bahamense, Karenia brevisulcata, Chaetoceros convolutus/concavicornis, Pfiesteria piscicida (the name piscimortis has been withdrawn). Correct Skeletonema costatum-complex [Zingone, A.,et al., J. Phycol. 41: 140–150]., Alexandrium tamarense-complex (the real tamarense is non-toxic). Add in category 4, Ostreopsis. You missed Gambierdiscus, Prorocentrum, Coolia
line 80. pelagophyte Aureumbra?
line 162. misspelling of Pseudo-nitzschia
Fig 2 and 3 presenting HAB cell abundances are meaningless without indicating which species is involved (see above; harm from cell-numbers is HAB species-specific). Analyse these data better.
line 186. except for
line 201. Skeletonema costatum complex. See Zingone papers, more than 1 species is involved
line 247. it is worthwhile pointing out. ...When referring to human poisonings, it helps to include data on toxin content of seafood
line 264. Alexandrium catenella in the Mediterranean almost certainly is group 4=A. pacificum
line 265-268. misspelling minutum
line 298. Thau lagoon has A. pacificum. The real A. tamarense is non-toxic; see John, U., et al. 2014. Formal revision of the Alexandrium tamarense species complex (Dinophyceae) taxonomy: the introduction of five species with emphasis on molecular-based (rDNA) classification. Protist 165, 779–804.
line 392. huxleyi
line 402. Greek waters have Karenia brevis like species, such as K. papilionacea, but the real Karenia brevis so far is confined to Florida. See also line 586
line 446. Gulf (with capital G when referring to a named location)
line 467. when referring to DSP, specify toxin levels
line 488. Chattonella globaosa should be Vicicitus globosus
line 543. Chattonella verruculosa should be Pseudochattonella
line 550 benefits for humans, which are essential for our well being
line 562. ciguatera is not discussed, but Gambierdiscus has been detected in Greek waters
The very poorly crafted section 6 on conclusions and perspectives does not reflect the present work, rather it brings up new material such as mitigation strategies tried elsewhere in the world, but this is highly incomplete and best not covered here. Focus on what the present work has contributed.
Some illustrations of key HAB species and coloured bloom waters would be useful.
Author Response
Reviewer 2
Comments and Suggestions for Authors
The value of this work is that it presents a comprehensive literature review (146 references) of Harmful Algal Bloom events in Mediterranean coastal waters, with a focus on Greek waters.
We would like to thank the reviewer for his/her positive disposition towards our work. All the comments and suggestions have been taken into consideration and the necessary changes have been highlighted in red in the revised manuscript to facilitate reading.
A limitation is that no effort is made to produce a quantitative analysis of trends (only Fig 1 on number of publications),
Thank you for your helpful comments. We have now included quantitative data according to your recommendations.
nor distribution maps presented (this is essential to include for non-Mediterranean readers; ideally all regions mentioned in the text should be labelled on a map),
We have now added maps in Figures 2 and 3 indicating the areas of the Mediterranean Sea and Greek Gulfs, respectively, where major HAB incidences have been documented. Especially for Figure 2, the map depicts the Mediterranean coastal areas with observed HABs, the abundance of main taxa involved, and the category of the HAB, as presented in Table 1.
nor the severity of economic impacts quantified.
It is true that it could be very useful if HAB studies included some sort of quantification of economic impacts. However, this is almost never easy to calculate, and since most of the literature included in this paper simply described the blooms and only some of them focused on the ecological impacts, we could not include in our review numeric values indicating economic losses.
The extensive claims of links with eutrophication are not quantified (line 18 in abstract; no nutrient data are presented!!);
Thank you for the suggestion, we have now included nutrient data where available to link increased eutrophication indicators in areas with observed frequent and severe HABs. Indicatively, please see L159-163, L179-180, L192-193, L203-204, L234-235, L276-277, L300-302, L349.
As admitted on line 67, these linkages are not always clear; note that only high biomass blooms (category 1 in Table 1; line 53) are commonly related to nutrients, but many other HABs such as Alexandrium can cause toxic problems at extremely low cell concentrations.
We have already mentioned that low abundance proliferations of certain taxa can cause adverse effects in the introduction L 46-48 & L 55-58. We have highlighted these parts in red.
There exist significant overlap with the excellent 2021 Zingone et al. Harmful Algae review (cited in ref 32; 501 HAEDAT events covered; 600 publications; 2350 Mediterranean records of 84 HAB species)), and this papers should attempt to differentiate itself eg., by the suggested modified title "Literature review of HABs in Mediterranean Sea, with a focus on Greek coastal waters". [there is no recent history presented; no time series analysed).
Thank you for the suggestion, we have now modified the title accordingly.
Analyse the literature data in Fig.1 better; break down into HAB categories; events in the 5 regions here discriminated etc; whether papers relate HAB to eutrophication etc.,
We appreciate this suggestion. We have thought a lot about how to improve this figure, which was included to show the increase of research on HABs in the Mediterranean area. However, we believe that any solution would either make the figure unnecessarily complicated or it would simply be incorrect. For example, it is impossible to break down the research papers according to HAB categories, since many of them fall into multiple categories, e.g. blooms with high biomass with water discoloration, along with toxin production harmful to wildlife and humans. Indeed, to increase the quantitative data presented in the review, we have now included a Mediterranean map in Figure 2, with quantitative data on major HAB events and separation of this events according to HAB category based on Table 1 (see also our reply to a previous comment). Also, we have added in text nutrient data where available.
line 30. ...marine environments, and can be expressions of ecosystem disturbance.
Thank you for the suggestion we have now modified it according to your suggestion
line 31. ..population live..
We apologize for the grammatical mistake; we have now corrected it.
This is not a global review, therefore all examples in Table 1 should be confined to Mediterranean waters. Delete Pyrodinium bahamense, Karenia brevisulcata, Chaetoceros convolutus/concavicornis, Pfiesteria piscicida (the name piscimortis has been withdrawn). Correct Skeletonema costatum-complex [Zingone, A.,et al., J. Phycol. 41: 140–150]., Alexandrium tamarense-complex (the real tamarense is non-toxic). Add in category 4, Ostreopsis. You missed Gambierdiscus, Prorocentrum, Coolia
Thank you for the suggestion we have now deleted Pyrodinium bahamense, Karenia brevisulcata, Chaetoceros convolutus/concavicornis, Pfiesteria piscicida, corrected Skeletonema costatum-complex, Alexandrium tamarense-complex, added Ostreopsis, Gambierdiscus, Coolia (Prorocentrum is already mentioned as Prorocentrum micans in category I & Prorocentrum lima in category II).
line 80. pelagophyte Aureumbra?
Aureococcus anophagefferens. It is now added.
line 162. misspelling of Pseudo-nitzschia
Corrected.
Fig 2 and 3 presenting HAB cell abundances are meaningless without indicating which species is involved (see above; harm from cell-numbers is HAB species-specific). Analyse these data better.
Thank you for your suggestion, we have now added the responsible taxa for the shown abundance on the figures.
line 186. except for
Corrected.
line 201. Skeletonema costatum complex. See Zingone papers, more than 1 species is involved
Thank you for the clarification, we have now replaced Skeletonema costatum with Skeletonema costatum complex
line 247. it is worthwhile pointing out. ...When referring to human poisonings, it helps to include data on toxin content of seafood
Unfortunately, no toxin data were presented in this case, but toxicity was inferred by mouse bioassays. Thus, we have changed the phrasing of this sentence to be more accurate.
line 264. Alexandrium catenella in the Mediterranean almost certainly is group 4=A. pacificum
Thank you for the clarification, we have now replaced A. catenella with A. pacificum throughout the text.
line 265-268. misspelling minutum
Corrected.
line 298. Thau lagoon has A. pacificum. The real A. tamarense is non-toxic; see John, U., et al. 2014. Formal revision of the Alexandrium tamarense species complex (Dinophyceae) taxonomy: the introduction of five species with emphasis on molecular-based (rDNA) classification. Protist 165, 779–804.
Thank you for the clarification. We have deleted A. tamarense in L298.
line 392. Huxleyi
Corrected.
line 402. Greek waters have Karenia brevis like species, such as K. papilionacea, but the real Karenia brevis so far is confined to Florida. See also line 586
Thank you for the clarification, we have now modified the text where necessary.
line 446. Gulf (with capital G when referring to a named location)
Corrected throughout the text.
line 467. when referring to DSP, specify toxin levels
The 2000 DSP outbreak in Greece was classified as such based on clinical symptoms characteristic for DSP, the common history of all patients reporting consumption of mussels, the absence of any known pathogen in faecal and blood samples from all hospitalized patients, no detection of antibodies to known pathogens, the subsequent analysis of mussels collected during the outbreak, and the detection of Dinophysis acuminata in the water of Thermaikos Gulf. In addition, the mouse bioassays performed during the outbreak were indicative of the presence of lipophilic toxins (Economou et al., 2007). We have now added this in the revised manuscript.
Since 2004 in Greece, testing of bivalves meat for lipophilic toxins is performed by liquid chromatography-mass spectrometry according to Regulation 853/2004 limits: for okadaic acid, dinophysistoxins and pectenotoxins together, 160 micrograms of okadaic acid equivalents per kilogram; for yessotoxins, 3.75 milligram of yessotoxin equivalent per kilogram; and for azaspiracids, 160 micrograms of azaspiracid equivalents per kilogram. In general levels of DSP toxins greater than 200 μg kg -1 in shellfish are considered dangerous for human consumption.
line 488. Chattonella globaosa should be Vicicitus globosus
Thank you for the clarification, we have now replaced Chattonella globaosa with Vicicitus globosus
line 543. Chattonella verruculosa should be Pseudochattonella
Thank you for the clarification, we have now replaced Chattonella verruculosa with Pseudochattonella
line 550 benefits for humans, which are essential for our well being
Thank you for the suggestion we have now modified it according to your suggestion.
line 562. ciguatera is not discussed, but Gambierdiscus has been detected in Greek waters
Thank you for the suggestion we have already mentioned Ciguatera Fish Poisoning along with the other HAB-related syndromes. We totally agree that CFP constitutes a novel and serious hazard for human health due to geographical expansion of Gambierdiscus in MS, and we have added both CFP and Gambierdiscus in Table 1.
The very poorly crafted section 6 on conclusions and perspectives does not reflect the present work, rather it brings up new material such as mitigation strategies tried elsewhere in the world, but this is highly incomplete and best not covered here. Focus on what the present work has contributed.
We have moved impacts of HABs to the introduction agreeing with the other reviewers’ suggestions. Thus, previous section 6 has become section 5. We have modified it by excluding mitigating measures that have been specifically applied outside the Mediterranean region, and we have added few lines (L489-493) mentioning European policies and mitigation strategies implemented by Mediterranean countries according to European legislation. However, it is true that mitigation strategies in the Mediterranean are not different from those applied everywhere in the world, so it was inevitable to mention general approaches that are being used to improve the quality of coastal waters.
Some illustrations of key HAB species and coloured bloom waters would be useful.
Thank you for the suggestion. We don’t have good quality original photos of key HAB species to include, however, the readers of our review would all be familiar with these taxa. On the other hand, we have modified Figure 2 to include a graphical representation of the observed HABs in each Mediterranean coastal area, the abundance of the main taxa involved, and the category of the HAB, as presented in Table 1.
Reviewer 3 Report
This study summarizes the current status of harmful algal blooms occurring in the Mediterranean coast. In particular, recent eutrophication by humans in many areas of the Mediterranean coast causes algal species related to algal blooms, their duration, and their effects on humans and ecosystems. etc. were arranged.
1. A detailed explanation of the species, density, and period of algal bloom occurring in each coast was good, but I hope that it will be expressed in a three-dimensional and spatial manner with maps rather than simply listing them.
2. The effect of harmful algae on humans and ecosystems is somewhat distant from the Mediterranean coast and is general content, so we hope to delete it.
3. It is judged that the conclusion and prospects need to be rewritten as there is a sense of distance from the outbreak of algae in the Mediterranean coast.
In conclusion, this review should be
1) Clearly re-establish the scope of research
2) Investigating the specificity of algal blooms in the Mediterranean regions rather than simply listing the current status (algal species, period of outbreak, environmental characteristics, etc.)
3) The cause of the algal bloom is tracked and differentiated from other regions. I strongly hope it will be rewritten.
Author Response
Reviewer 3
Comments and Suggestions for Authors
This study summarizes the current status of harmful algal blooms occurring in the Mediterranean coast. In particular, recent eutrophication by humans in many areas of the Mediterranean coast causes algal species related to algal blooms, their duration, and their effects on humans and ecosystems. etc. were arranged.
We would like to thank the reviewer for his/her comments and suggestions. We read carefully all the comments and changes were made accordingly and in agreement of the other reviewers’ recommendations. All changes in the revised manuscript are highlighted in red to facilitate reading.
- A detailed explanation of the species, density, and period of algal bloom occurring in each coast was good, but I hope that it will be expressed in a three-dimensional and spatial manner with maps rather than simply listing them.
Thank you for the suggestion. We have now modified Figure 2 to include a map of the Mediterranean region with graphical representations of the observed HABs in each Mediterranean coastal area, the abundance of the main taxa involved, and the category of the HAB, as presented in Table 1.
The effect of harmful algae on humans and ecosystems is somewhat distant from the Mediterranean coast and is general content, so we hope to delete it.
We have now moved this part to the introduction, which includes more general statements, and we believe is better suited, as was also the recommendation of another reviewer.
It is judged that the conclusion and prospects need to be rewritten as there is a sense of distance from the outbreak of algae in the Mediterranean coast.
We have modified this section by excluding mitigating measures that have been specifically applied outside the Mediterranean region, and we have added few lines (L489-493) mentioning European policies and mitigation strategies implemented by Mediterranean countries according to European legislation. However, it is true that mitigation strategies in the Mediterranean are not different from those applied everywhere in the world, so it was inevitable to mention general approaches that are being used to improve the quality of coastal waters.
In conclusion, this review should be
1) Clearly re-establish the scope of research
The title of the paper has been modified based also on another reviewer’s comments, to emphasize that the paper regards a literature review of HABs in the Mediterranean coastal waters with specific focus in the Greek coasts. Also, the aim of the review has been modified accordingly.
2) Investigating the specificity of algal blooms in the Mediterranean regions rather than simply listing the current status (algal species, period of outbreak, environmental characteristics, etc.)
With respect to the reviewer, we believe that this part has been accomplished in the original manuscript. Throughout the text we describe the causative algal species, abundance range, period of outbreak, and we do not focus on the current status (which in many cases there are no published information).
Furthermore, in the revised manuscript we have added nutrient data (Total Nitrogen and Total Phosphorus, which are directly associated with eutrophication) where available to link increased eutrophication indicators in areas with observed frequent and severe HABs. Indicatively, please see L159-163, L179-180, L192-193, L203-204, L234-235, L276-277, L300-302, L349.
3) The cause of the algal bloom is tracked and differentiated from other regions. I strongly hope it will be rewritten.
Thank you for the suggestion. We have now added additional quantitative information on nutrient data where available (please see our response to the previous comment), attempting to link in each HAB case, the eutrophication status of the area at the time of the outbreak with the bloom. This is not easy for all cases, as environmental data are missing or are incomplete for many research papers.
the rebuttal for reviewer 3 in second round is attached below.

Round 2
Reviewer 2 Report
I am happy with the authors' responses and improvements to reviewer comments. I notably like the new figs 2 and 3
There are still some minor corrections needed.
Line 48. "which can be poisonous for shellfish".. should be "which can cause shellfish toxicity even in low concentrations"... [ algal toxins are rarely harmful to shellfish themselves, but dangerous for humans, birds etc with complex nervous systems]
Fig.2. bottom "Chattonella" is misspelled twice [named after the French scientist Chatton, with double t]
line 476. "verruculosa" [double r]
Author Response
I am happy with the authors' responses and improvements to reviewer comments. I notably like the new figs 2 and 3
There are still some minor corrections needed.
We would like to thank the reviewer for the helpful comments and suggestions that improved our original manuscript. All new corrections are now included in the new version.
Line 48. "which can be poisonous for shellfish".. should be "which can cause shellfish toxicity even in low concentrations"... [ algal toxins are rarely harmful to shellfish themselves, but dangerous for humans, birds etc with complex nervous systems]
Modified according to the suggestion.
Fig.2. bottom "Chattonella" is misspelled twice [named after the French scientist Chatton, with double t]
We apologize for the misspelling. We have now corrected it.
line 476. "verruculosa" [double r]
We apologize for the misspelling. We have now corrected it.
Reviewer 3 Report
The authors did their best to partially respond to the first comment and proceeded with the revision of the manuscript.
1. In particular, in Figures 2 and 3, HAB target species and extant amounts are indicated on the map, and fig.3 also wants to indicate biomass like fig.2.
2. However, the general description of the properties (toxins and effects) of HAB mentioned in the introduction should be minimized and replaced with references.
3. The suggestion on HAB generation suppression and control method in the conclusion part is not very fresh. In particular, the method for blocking physicochemical pollution sources only in the Greek waters (coast) does not show any differentiation.
4. Lastly, this review has a limitation that the HAB occurrence characteristics of the Greek coast are not differentiated from other regions. It is necessary to discover the characteristics of the coast of Greece.
Author Response
The authors did their best to partially respond to the first comment and proceeded with the revision of the manuscript.
1. In particular, in Figures 2 and 3, HAB target species and extant amounts are indicated on the map, and fig.3 also wants to indicate biomass like fig.2.
We thank the reviewer for the positive disposition towards our revised manuscript and the helpful comments that improved our original review paper.
2. However, the general description of the properties (toxins and effects) of HAB mentioned in the introduction should be minimized and replaced with references.
Thank you for the suggestion, however, in our opinion, this part of the introduction is essential to introduce the reader to HABs of categories II, III, and IV (Table 1), which are frequent and ubiquitous at the Mediterranean coasts. We believe that this short description of the properties of HABs help the readers understand the significant impacts of the HABs presented on following chapters, on human, wildlife, and ecosystem health.
3. The suggestion on HAB generation suppression and control method in the conclusion part is not very fresh. In particular, the method for blocking physicochemical pollution sources only in the Greek waters (coast) does not show any differentiation.
It is true that mitigation measures throughout the world are similar and generic. However, these measures are used worldwide, and of course in Greece following international guidelines and literature, as mentioned indicatively in L489-493. Furthermore, the main purpose of the paper is to review and describe HAB events, and not proposing mitigation actions.
4. Lastly, this review has a limitation that the HAB occurrence characteristics of the Greek coast are not differentiated from other regions. It is necessary to discover the characteristics of the coast of Greece.
With respect to the reviewer we do not agree with this comment. In every sub-chapter we describe in detail the unique characteristics of each area. For Greek gulfs, please see L381-388 for Saronikos Gulf, L399-408 for Thermaikos Gulf, L440-442 for Evoikos Gulf, L450-453 for Pagasitikos Gulf, L458-460 for Kavala Gulf, L467-470 for Maliakos Gulf, and L472-475 for Amvrakikos Gulf.